# Living with Risk, Aging with Uncertainty: A Narrative Review of Health and Genetic Vulnerability in Huntington’s Disease

**DOI:** 10.3390/biomedicines13102498

**Published:** 2025-10-14

**Authors:** Adriana V. Muñoz-Ortega, David Conde Caballero, Lorenzo Mariano Juárez

**Affiliations:** 1Department of Computer and Telematics Engineering, School of Engineering, University of Extremadura, 10003 Cáceres, Spain; 2Interdisciplinary Group for Studies in Society, Culture and Health, University of Extremadura, 10003 Cáceres, Spain; dcondecab@unex.es (D.C.C.); lorenmariano@unex.es (L.M.J.); 3Department of Nursing, Faculty of Nursing and Occupational Therapy, University of Extremadura, 10003 Cáceres, Spain; 4Department of Psychology and Anthropology, Faculty of Teacher Training, University of Extremadura, 10003 Cáceres, Spain

**Keywords:** genetic predisposition to disease, coping strategies, Huntington’s disease, genetic counselling, psychosocial factors, ethics, aging

## Abstract

**Background/Objectives**: Huntington’s disease (HD) is an autosomal dominant, neurodegenerative disorder that, because of the availability of presymptomatic genetic testing, places at-risk individuals in an anticipatory situation of great emotional, ethical, and social complexity. This review synthesizes the subjective experiences and coping strategies of individuals aware of their genetic risk before clinical diagnosis, emphasizing the importance of patient and family narratives as critical sources of evidence for enhancing care protocols. **Methods**: This work is a narrative review supported by a systematic literature search. Of the 75 studies analyzed, 22 met the inclusion criteria—i.e., qualitative research, reviews, and case studies addressing emotional, cognitive, behavioral, and ethical coping mechanisms. The information was structured within a thematic matrix, and inductive coding was applied to identify recurring patterns, unresolved tensions, and gaps in the literature. **Results**: Presymptomatic genetic diagnosis may trigger processes of anticipatory grief, disrupt individual and familial identity, and lead to constant somatic self-monitoring. Coping strategies vary from proactive approaches—e.g., seeking information and building support networks—to narrative reframing that emphasizes acceptance and the resignification of risk. Analyzing these narratives allowed us to identify silenced ethical dilemmas and family rituals that help alleviate uncertainty—dimensions often overlooked by traditional quantitative methods. Moreover, risk awareness also impacts reproductive and care planning decisions, underscoring the importance of ongoing, context-sensitive support. **Conclusions**: Coping with genetic risk in Huntington’s disease extends beyond the biomedical aspects to encompass relational, ethical, and narrative dimensions. Incorporating narrative-based medicine into genetic and psychosocial counseling is crucial for identifying implicit needs and providing more empathetic, individualized care.

## 1. Introduction

Huntington’s disease (HD) is an autosomal dominant, neurodegenerative disorder caused by an abnormal CAG (cytosine–adenine–guanine) repeat expansion in the huntingtin (HTT) gene [1]. This results in the pathological production of mutant huntingtin protein, triggering a cascade of neurotoxic processes and progressive neuronal dysfunction [2,3,4]. A recent meta-analysis reported a pooled prevalence of 6.37 cases per 100,000 inhabitants in Europe, although noting geographical variations attributable to differences in diagnostic and registration methods [5,6].

Clinically, HD presents as a combination of motor, cognitive, and psychiatric symptoms, following a progressive course that significantly diminishes patients’ quality of life and increasingly destabilizes their familial support systems [7,8,9]. As the disease advances, patients require greater assistance with both basic and instrumental activities of daily living. This escalating care burden primarily falls on family members, imposing physical, emotional, and financial strains on caregivers [7,10]. Some studies have highlighted both the subjective burden experienced by families and the objectively measurable impact on caregiver wellbeing [11,12]. The economic impact of HD is considerable. In the United States, total annual direct and indirect costs are estimated at €101,971 per patient, highlighting the substantial financial and social burden on both families and healthcare systems [13]. Direct costs alone represent a considerable proportion of this total [14].

Its autosomal dominant inheritance pattern allows for the distinction of two psychosocial phases in the course of the disease. The first involves the preclinical stage, during which individuals are aware of their potential genetic risk and must consider whether to undergo presymptomatic genetic testing [15]. The second phase follows diagnostic confirmation or the onset of clinical symptoms, marking the transition to manifest disease [16]. 

This distinction stems from the availability of presymptomatic genetic testing, which confers HD a unique characteristic [17]. In 1993, the discovery of the CAG repeat expansion in the HTT gene as the cause of HD paved the way [18,19] for presymptomatic genetic testing in asymptomatic individuals carrying the mutant allele. The standard method involves analyzing a 5–10 mL ethylenediaminetetraacetic acid (EDTA)-treated blood sample using polymerase chain reaction (PCR) followed by electrophoresis to quantify the number of CAG repeats in exon 1 of the HTT gene [20]. A repeat count of ≥40 is associated with a fully penetrant disorder [21]. This approach achieves a mutation detection sensitivity of 100% [22]. The clinical protocol followed includes several phases: pre-test consultation with genetic and psychological counseling, sample collection, molecular analysis, and post-test sessions to communicate results and provide follow-up support [23,24].

The availability of genetic diagnosis has profound emotional [25], ethical, and social implications, as it defines a context marked by uncertainty, anticipatory grief, and complex decision-making [14]. Individuals at risk for HD face the possibility of a positive diagnosis and must make decisions regarding their future, reproductive planning, and whether to undergo presymptomatic testing [26]. Recent studies reported high levels of anxiety, depression, and stress in both asymptomatic carriers and non-carriers, highlighting the frequent need for psychological support following genetic testing [25,27]. In this context, coping strategies are critical. Research by Hubers et al. [28] demonstrated that individuals in the prodromal phase, as well as their partners, employed coping mechanisms such as acceptance, emotional support, and active planning.

However, despite its significance, knowledge of how genetic risk is managed prior to clinical diagnosis remains limited. Various studies have demonstrated that coping skills significantly impact the quality of life and adherence to medical recommendations, both of which are crucial for the optimal management of the disease [29,30,31]. In this context, it is essential to examine how individuals at risk for HD interpret and navigate their experience—not only from a clinical or psychological viewpoint, but also through their personal narratives and interpretive frameworks. 

The qualitative analysis of experiential narratives provides a valuable approach for accessing these subjective dimensions, revealing not only symptomatology and disease progression but also the influence of genetic information on the identity and future life planning of the individuals affected [32]. From a narrative-based medicine perspective, analyzing experiential accounts helps identify and address ethical conflicts, family dilemmas, and early coping strategies that are often overlooked or remain invisible in quantitative data [33]. Similarly, classic literature emphasizes that actively listening to patients’ stories improves the doctor–patient relationship and enhances therapeutic adherence by fostering a more empathetic approach to healthcare [34]. The accounts of patients and their families thus constitute valuable evidence that may inform protocol development and help reduce persistent uncertainties [35,36]. Medical anthropology has suggested broadening the definition of “evidence” in healthcare to include belief systems, cultural values, and care practices that emerge through patients’ narratives [37]. Narrative-based medicine—understood as the systematic collection and analysis of patients’ experiential accounts—allows for the capture of nuanced aspects of the anticipatory experience (including fears, hopes, and mitigation rituals) that are inaccessible to standardized instruments, and offers valuable insights for designing more context-sensitive interventions [33,34,38].

This study aims to examine the available evidence on coping experiences associated with knowledge of genetic risk for HD before clinical diagnosis. It seeks to integrate the narratives and strategies employed by at-risk individuals and their close relatives to understand the emotional and social impact of the anticipatory experience, as well as the resources mobilized to facilitate progressive adaptation at both the individual and family levels. A deeper understanding of these experiences is essential for informing clinical practice and psychosocial support, enabling more sensitive and personalized interventions aligned with the real needs of people living with the fear of a positive diagnosis. 

## 2. Materials and Methods

### 2.1. Research Design

This study is based on a narrative review of coping with genetic risk in Huntington’s disease, with an emphasis on the experiences of patients and their relatives. Although not a full systematic review, we employed a structured search strategy and applied elements of the PRISMA 2020 framework (Preferred Reporting Items for Systematic Reviews and Meta-Analyses; Page et al., 2021; University of Oxford, Oxford, UK) (e.g., flow diagram, reporting of inclusion/exclusion criteria) to enhance methodological transparency [39]. This hybrid approach allowed us to combine the breadth of a narrative review with the methodological rigor of systematic searching, offering an updated overview of the psychosocial impact and the clinical and ethical implications of genetic risk in this neurodegenerative disorder, following the recommendations of Greenhalgh et al. [40].

### 2.2. Literature Search Strategy

The literature search was conducted between January and April 2025 across five internationally recognized, online scientific databases: PubMed (U.S. National Library of Medicine, Bethesda, MD, USA), Scopus (Elsevier B.V., Amsterdam, The Netherlands), Web of Science (Clarivate, Philadelphia, PA, USA), PsycINFO (American Psychological Association, Washington, DC, USA) and EMBASE (Elsevier B.V., Amsterdam, The Netherlands), in order to ensure broad coverage of both psychosocial and biomedical literature. Boolean combinations of keywords in both English and Spanish were used, including the following terms and their equivalents in Spanish: “genetic risk,” “coping strategies,” “Huntington’s disease,” “genetic counseling,” “psychosocial impact,” and “ethical issues”. This broad search strategy was designed to capture the widest possible range of studies at the intersection of clinical, ethical and psychosocial perspectives. Additionally, to capture transferable psychosocial evidence relevant to anticipatory genetic risk, we considered studies addressing analogous diagnostic experiences (e.g., receipt of a genetic psychiatric, or rare-disease diagnosis) when they offered clear insights into identity reconfiguration, diagnosis uncertainty, family communication, or coping strategies. Such inclusions were intended to enrich the thematic synthesis rather than to substitute disease-specific biomedical evidence. 

In line with the PRISMA 2020 framework [39], the research question was formulated to explore how individuals at genetic risk of Huntington’s disease (patients and relatives) cope with genetic risk, focusing on coping strategies, psychosocial impact, and the ethical implications of predictive genetic testing in clinical, counselling and community settings.

Filters were applied to limit the results to peer-reviewed articles and full-texts available in English, Spanish, and Portuguese, covering the period from January 2000 to September 2025.

### 2.3. Inclusion and Exclusion Criteria

#### 2.3.1. Inclusion Criteria

Studies were eligible for inclusion if the met the following criteria: (1) addressed genetic risk in Huntington’s Disease (HD) as a central focus; (2) reported on theoretical studies, systematic reviews, narrative analyses, qualitative research, or case studies focused on individuals at risk for HD (patients or relatives); (3) explored at least one of the following dimensions: of emotional, cognitive, or behavioral coping with genetic risk; communication of presymptomatic diagnosis; ethical implications of predictive testing; or models of genetic counseling; (4) were peer-reviewed articles, published in English, Spanish or Portuguese between January 2000 and September 2025. 

In addition, we included non-HD studies when they provided transferable psychosocial insights directly relevant to anticipatory genetic risk (e.g., diagnostic uncertainty, family communication, or identity reconfiguration). These studies were retained only if their applicability to the HD context was explicit and clearly documented in the extraction matrix. 

#### 2.3.2. Exclusion Criteria

Studies were excluded if they: (1) focused exclusively on molecular, genetic or laboratory aspects without a human, clinical or psychosocial component, (2) were duplicated studies, unindexed publications, or not peer-reviewed. 

### 2.4. Literature Selection and Analysis

The selection of studies was conducted in two stages: (1) an initial screening of titles and abstracts, and (2) a full-text review of potentially eligible articles. The screening was performed by the first author and independently verified by two additional members of the research team. Any discrepancies in article inclusion or interpretation were resolved through discussion until consensus was reached, ensuring methodological rigor and transparency.

During full-text review, non-HD papers were assessed for transferability using predefined considerations (similarity of diagnostic uncertainty, relevance of the psychosocial construct, and applicability to genetic risk scenarios). Only studies that met these transferability criteria were included in the thematic synthesis. All such inclusions are explicitly indicated in the data extraction table. 

Relevant information was extracted and organized into a structured thematic matrix, which included information on study design, participants, country/region, methods, and key findings. This method facilitated the identification of recurring patterns, areas of consensus, unresolved tensions, and gaps in the existing literature. 

The synthesis was guided by a thematic analysis approach, with particular attention to the following dimensions: emotional and behavioral coping with genetic risk, genetic counselling models, and ethical dilemmas associated with presymptomatic diagnosis. Following this matrix, we elaborated a thematic narrative analysis of the results. 

Triangulation of perspectives among team members further strengthened the validity of the findings and supported a multidimensional understanding of genetic risk management in HD (see Figure 1).

### 2.5. Ethical Considerations

As this narrative review did not involve direct participation of individuals or the use of identifiable clinical data, it did not require prior approval from an ethics committee, in line with current research legislation in Spain.

## 3. Results

A total of 75 articles were analyzed, of which 22 met the final inclusion criteria for this narrative review. The results were organized into four broad thematic categories encompassing the main dimensions of coping with genetic risk in HD: (1) The impact of genetic diagnosis; (2) reconfiguration of identity and family dynamics; (3) emotional management and coping strategies; and (4) future life plans, trajectories and advance planning. Table 1 presents an overview of the studies included in the review, while the following section provides more detailed narrative results.

### 3.1. The Impact of Genetic Diagnosis

Accurate genetic diagnosis informs both clinical and reproductive decision-making, facilitates access to psychosocial support resources, and reduces uncertainty by validating family experiences associated with the disease. Consequently, it represents a pivotal moment in the experiential journey of patients and their families. In families affected by HD, where the intergenerational impact is particularly profound, the diagnosis assumes a significance that extends beyond its purely biomedical value. Individuals seek not only a scientific explanation but also an interpretive framework to understand and reframe their experiences, make informed decisions about their lives, and anticipate future care needs. In this context, genetic testing serves as a tool that integrates biological, identitary, emotional, and social dimensions, providing meaning to a reality often perceived as uncertain or ambiguous.

The structured analysis of narratives extracted from the studies included in this review reveals critical dimensions beyond conventional quantitative findings, including unspoken ethical dilemmas, early coping patterns, and family rituals aimed at mitigating uncertainty [41]. Furthermore, incorporating the voices of HD patients enhances the interpretation of clinical and psychosocial outcomes by validating the subjective dimension of coping with genetic risk. These findings underscore the need to systematically integrate narrative-based medicine into genetic and psychosocial support protocols.

Negative diagnostic test results, although generally a source of relief, can sometimes lead to feelings of survivor guilt, identity dissonance, and conflicts in the reconstruction of family roles [42]. Costa et al. [41] emphasize that these reactions are particularly pronounced in sociocultural contexts with strong family and community ties—such as certain Asian cultures—where voluntary refusal of genetic testing is often explained by its potential to disrupt the emotional and relational balance, threatening group cohesion and triggering stigmatization, discrimination, or overprotection.

Individuals who receive negative test results often face the emotional challenge of integrating into families affected by HD and may, consciously or unconsciously, experience a form of “relational loss” by distancing themselves from family members who carry the mutation [43].

The concept of “genomic designation”—i.e., classifying carriers of pathogenic variants as “diseased” before the onset of clinical symptoms—has profoundly transformed the experience of genetic diagnosis [44]. By shifting the diagnostic threshold to presymptomatic stages, the individual experience is fundamentally transformed: receiving a result ceases to be merely molecular data; instead, it becomes a life-changing event that initiates an ongoing process of reassessing one’s health and future expectations. 

This new diagnostic paradigm facilitates earlier preventive interventions with the potential to alter the disease trajectory; however, it also introduces complex psychosocial challenges. Individuals receiving a presymptomatic diagnosis often report feelings of anticipatory anxiety and a heightened sense of responsibility toward their descendants, alongside an increasing reliance on the healthcare system [45]. Studies, such as that of Quaid et al. [46], show how the “at-risk” identity can become inextricably woven into personal narratives, shaping decisions related to work, family, and social life, so that the impact of a presymptomatic diagnosis persists long after testing. 

In summary, the impacts of genetic diagnosis in HD are multifaceted and far-reaching, extending beyond the individual to influence family dynamics, the healthcare system, and the wider communities and social systems in which these individuals are part. Acknowledging this multidimensionality is crucial for developing care models that incorporate emotional support, ethical guidance, and a strong respect for individual autonomy. 

### 3.2. Reconfiguring Identity and Navigating Family Dynamics

Awareness of an individual’s genetic status often prompts a reorganization of their personal and existential identity. While it offers a lens through which to reinterpret symptoms and familial experiences, it may also initiate processes of anticipatory grief and a sense of loss projected into the future [44,47,48]. Genetic certainty introduces a biographical disruption that compels individuals to reassess life expectations, personal goals, and prior notions of normality. In this context, many at-risk individuals report experiencing a distinct “before” and “after” the diagnosis, even in the absence of clinical symptoms, accompanied by a profound sense of existential vulnerability that is often difficult to articulate or share. 

HD gene carriers in the presymptomatic phase of the disease may experience heightened somatic self-monitoring, interpreting even minor physical changes as potential early indicators of disease onset. As Mahmood et al. [49] suggested, this experience alters the perception of both body and time, fostering a hyperawareness and constant monitoring that may affect psychological wellbeing. The relationship with the body undergoes a process of “somatic reassessment,” where everyday gestures, lapses in memory, and routine movements are scrutinized through the lens of genetic suspicion. This heightened vigilance might cause anxiety and result in avoidance behaviors, social withdrawal, or premature medical interventions, even during presymptomatic phases. 

Furthermore, genetic testing and its relational context are profoundly shaped by family ties. Living with individuals affected by HD in various ways creates an emotional capital that impacts life decisions and shapes intergenerational relationships [50]. The experience of risk is rarely solitary; instead, it is often closely intertwined with narratives of care, family commitments, shared fears, and unspoken agreements. Within this context, phenomena such as overprotection, information concealment, and conflicts over autonomy [51] frequently emerge, as noted by Cesanelli & Marguiles [52], Oliveira et al. [53], and Pleutim et al. [54].

The interpretation of genetic risk is shaped by an individual’s life trajectory, beliefs, and family history. As Anderson et al. [55] and Etchegary [45] suggest, the test alone does not reduce uncertainty unless it is framed within a coherent personal narrative. Indeed, the meaning attributed to risk is shaped not only by objective data but also by an individual’s capacity to integrate it into their identity and personal worldview. Factors such as age, parenthood, experiences of loss or caregiving, and cultural values influence how risk is experienced and woven into an individual’s life narrative. Thus, for some people, knowing their genetic status may be empowering, while for others, it may represent an emotional burden that challenges their psychological stability and affective relationships. 

### 3.3. Emotional Management and Coping Strategies

Strategies focused on primary control enable individuals to maintain a degree of self-determination. These include proactively seeking information, participating in support networks, developing emotional regulation skills, and engaging in meaningful activities [46,56]. These actions enable a greater sense of personal control, even in the face of clinical uncertainty, and can serve as protective mechanisms against psychological distress. 

Other strategies center on acceptance, comparison with other illnesses, spirituality, activism, and integrating genetic risk into one’s life narrative. According to Etchegary [57] and Quaid et al. [46], these forms of secondary coping can enhance psychological resilience by offering a symbolic framework through which the threat of disease can be reinterpreted. From this perspective, genetic risk is not viewed solely as a threat, but also as an opportunity to reflect on personal values, relationships, and priorities. 

In contrast, anticipation of the future may be addressed by either deliberate avoidance or active engagement with the present. Wieringa et al. [58] suggested the concept of “biographical disruption” to describe how individuals reconstructed their life expectations in response to genetic risk, integrating forms of hope rooted in science, family legacy, or personal coping capacities. This process of reconstruction may entail both a redefinition of life plans and a reappropriation of everyday life through small, meaningful actions. 

Wexler [59] argued that genetic certainty does not necessarily bring peace of mind but can instead generate an existential void. Her work further examined how risk communication unfolded as a dynamic and ambivalent process, marked by fluctuations between the desire for knowledge and the fear of definitive answers. This complexity has been further supported by subsequent studies [58], which emphasize that the coping process is rarely linear or predictable and is shaped by personal, familiar and contextual factors.

### 3.4. Life Plans, Trajectories and Advance Planning

Awareness of genetic risk alters the perception of time, which may feel denser, accelerated, or even threatening. This phenomenon often manifests in a constant monitoring of the body for potential symptoms, generating a sense of urgency and anticipatory anxiety [58]. As a result, the sense of time becomes altered, leading many individuals to reorganize their personal, educational, and professional goals in light of a future shaped by the possibility of developing the disease. 

Experiences with family members who are affected often motivate individuals to engage in advance care planning (ACP). Rasmussen & Alonso [60] and Ekkel et al. [61] identify various strategies, including explicit planning, avoidance, and delegating decisions to close, trusted individuals. These choices are rarely purely rational; instead, they are often emotionally charged and shaped by family loyalties and implicit expectations.

Regarding reproductive choices, Aguilar-Caro & Campo Oviedo [62] and Rubio Vizcaya [50] show that genetic counseling must carefully balance cultural, religious, and family values, particularly when discussing options such as preimplantation genetic diagnosis. Final decisions involve intimate negotiations between the couple, their relatives, and the regulatory and ethical frameworks that govern access to reproductive technologies. Consequently, the dilemma of whether to pass on the genetic risk becomes a profoundly existential question. 

To synthesize the findings and provide a structured representation of the analytical categories identified, we developed a conceptual model (Figure 2). This model illustrates the dynamic process through which individuals at genetic risk for Huntington’s disease navigate risk awareness, identity work, coping strategies, and planning decisions. It also highlights the moderating role of contextual factors such as culture, family roles, prior caregiving experience, and the health system. By integrating these dimensions, the model offers a comprehensive framework that connects the themes emerging from the reviewed studies and clarifies their interrelations, serving as a bridge between the empirical results and the subsequent discussion.

## 4. Discussion

This narrative review provides an in-depth exploration of the multifaceted dimensions that influence the experience of living with a genetic risk for Huntington’s disease. By organizing the findings into four analytical categories, the review highlights both the clinical and personal implications of the genetic diagnosis, as well as the identity, familiar, and emotional impacts associated with it. 

Firstly, genetic diagnosis in HD is perceived as an ambivalent tool. While it can offer diagnostic clarity and support clinical and reproductive decision-making [44], its potential to provide reassurance and alleviate anxiety is shaped by the individual’s cultural, familial, and biographical context. Particularly noteworthy is the finding that even a negative result can elicit feelings of guilt, identity dissonance, and challenges in reconfiguring family roles, instead of providing relief [42].

The identity of HD gene carriers is reshaped by the genomic designation, which anticipates a disease that has yet to manifest clinically, thereby transforming the affected individuals’ relationships with their body, their sense of time, and their interpersonal connections. As noted by Navon [44], genomic designation introduces an anticipatory temporality that impacts the individual’s biography. Research has shown that genetic testing not only provides biomedical knowledge but also gives rise to subjective meanings that may be experienced as a form of anticipatory suffering [47,48,58].

In this context, identity emerges as a central factor in coping with genetic risk. The experience of being “at risk” is internalized both physically—through constant bodily self—monitoring—and relationally, within complex family dynamics marked by concealment, overprotection, and tensions around autonomy [53,54,63]. Similarly, the experience of risk is shaped by what Etchegary [45] terms “zones of relevance”—where personal, historical, and cultural factors influence how the test is interpreted and integrated into an individual’s life narrative.

From a coping perspective, individuals employ a range of strategies, from active control of their environments to symbolic reframing of the threat. Some engage in emotional regulation, information seeking, or participation in support networks [46,56], while others draw on spiritual, narrative, or activist approaches to make sense of their experience [51,58]. In all cases, managing emotional risk emerges as a non-linear process, profoundly shaped by the ambivalence inherent in genetic knowledge, as described by Wexler [59] and Wieringa et al. [58].

Additionally, recent reviews [64,65] underscore the value of experiential narratives in highlighting inequalities in access to genetic testing and informing family support policies, thereby establishing narrative-based medicine as a key element in the comprehensive management of HD. However, evidence also indicates that psychological support interventions for HD remain scarce, fragmented, and insufficiently tailored to the complex needs of patients and families, despite their documented demand for such resources [66].

Finally, awareness of genetic risk prompts many individuals to reassess their life plans, from advance care planning to reproductive decisions. This altered perception of biographical time creates a sense of urgency that underpins the development of long-term strategies for both themselves and future generations [27,60,61]. Recent evidence further shows that while asymptomatic carriers are generally open to receiving predictive information for planning purposes, they are more hesitant about progression predictions due to concerns about psychological burden and uncertainty [67]. This forward-looking dimension of coping underscores how genetic knowledge reshapes not only people’s relationship with the present but also with their future, their heritage, and their descendants.

In summary, the findings of this review demonstrate that coping with genetic risk in HD is a multidimensional process extending beyond the biomedical sphere to encompass psychological, familial, cultural, and existential issues. This complexity underscores the importance of interdisciplinary approaches and comprehensive psychosocial care to support genetic testing and monitoring, respecting each individual’s unique life story.

Understanding the narratives and lived experiences of people facing genetic risk is essential for grasping the meanings they assign to the disease, the diagnosis, and their life choices. These narratives reveal subjective dimensions—such as fear, guilt, hope, and ambivalence—that remain inaccessible through purely biomedical or quantitative methods. Integrating these voices enhances our understanding of the psychosocial impact of genetic risk and informs the development of more empathetic, culturally sensitive, and person-centered interventions. Moreover, bringing these experiences to light challenges normative models of coping and acknowledges the diverse trajectories that may unfold across varying family, social, and healthcare contexts. 

### Limitations

This study has several limitations that warrant consideration. First, although we applied elements of the PRISMA 2020 framework to enhance transparency, this remains a narrative review rather than a fully systematic review, which may limit the comprehensiveness and reproducibility of our findings. However, the narrative format also allowed for the integration of qualitative and experiential evidence that is often excluded from systematic reviews, thereby enriching the scope of the synthesis.

Second, despite our efforts to ensure broad database coverage, we were unable to access the EMBASE database, as none of the participating institutions hold subscriptions to this resource. Although we recognize EMBASE’s importance in capturing biomedical and pharmaceutical literature in health sciences, this limitation may have resulted in the exclusion of relevant studies indexed exclusively in that database. Nonetheless, the inclusion of PubMed, Scopus, Web of Science, and PsycINFO provided substantial coverage of both biomedical and psychosocial domains.

Third, the restriction to articles published in English, Spanish and Portuguese may have excluded relevant studies in other languages, potentially limiting the cultural and geographical representativeness of our findings. Yet, given that most publications on Huntington’s disease are produced in English, the overall impact of this restriction is likely to be limited.

Finally, the thematic synthesis approach, while valuable for identifying patterns and conceptual frameworks is inherently interpretative and may be influenced by the research team’s disciplinary perspective. To mitigate this, triangulation among team members was applied, enhancing the credibility and robustness of the synthesis. Despite this limitation, this review provides comprehensive insights into coping with genetic risk in Huntington’s disease and establishes a foundation for future systematic investigations.

## 5. Conclusions

The experience of genetic risk in HD is profoundly transformative—characterized by fear, uncertainty, and the need to reassess and reframe both individual and familial identity. Beyond clinical and diagnostic parameters, individuals at risk must face an anticipatory dimension of the disease that shapes their life trajectories well before symptom onset. Pre-diagnosis coping strategies should not be viewed as isolated responses but as relational, ethical, and emotional processes where family history, future life plans, and present identity converge. 

The findings of this review highlight the limitations of traditional biomedical models in addressing the psychosocial complexity inherent in genetic risk. In light of this limitation, the narratives of patients and family members emerge as essential sources of knowledge—revealing the profoundly subjective and experiential dimensions of suffering, resistance, ambivalence, and agency, and providing crucial insights for developing more humane, sustainable, and context-sensitive interventions. Narrative-based medicine is not merely a complementary methodology but a vital tool for listening to issues that quantitative data alone cannot convey. 

Despite advances in genetic counseling and the development of psychoeducational resources, significant gaps remain in the comprehensive care of asymptomatic individuals at risk for HD—including inequalities in access to genetic testing, a lack of protocols sensitive to presymptomatic phases, limited standardization of best clinical practices, and a shortage of longitudinal studies examining changes in risk experiences. These gaps undermine not only the quality of care provided to at-risk individuals but also the full exercise of their autonomy. 

It is therefore imperative to approach genetic risk through a narrative and context-sensitive lens, focusing on the lived experiences of the individuals facing the inherited threat. Understanding how risk is experienced transcends academic or clinical concerns—it is a responsibility owed to those who, although currently healthy, carry the heavy burden of a future inscribed in their genes. 

## Figures and Tables

**Figure 1 biomedicines-13-02498-f001:**
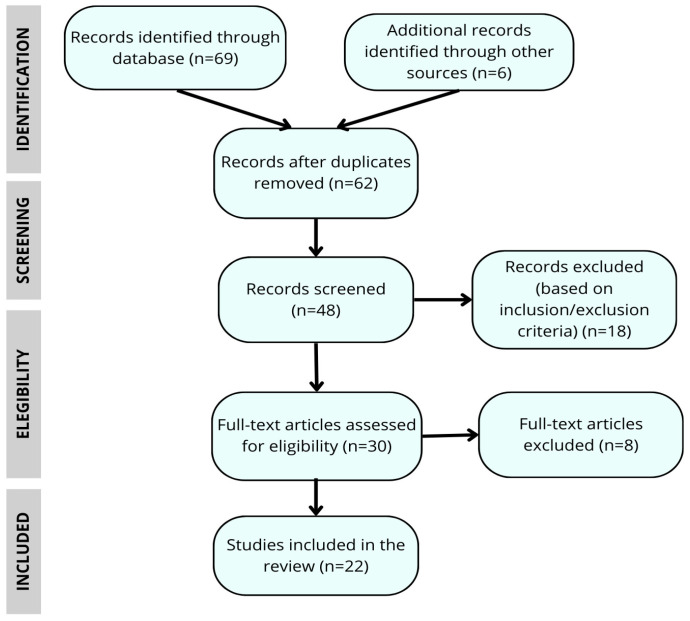
Flow chart of the article selection process for the narrative review.

**Figure 2 biomedicines-13-02498-f002:**
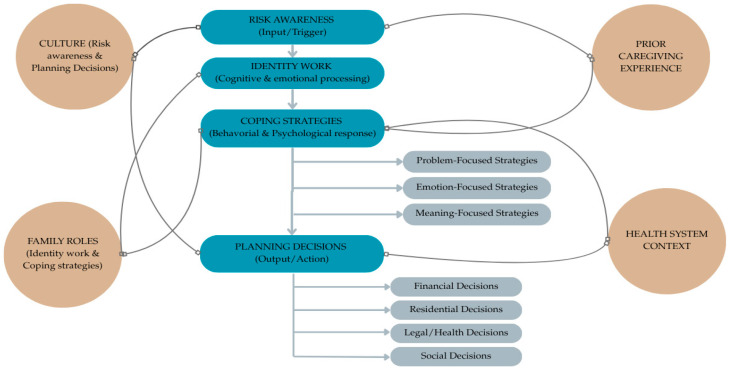
Conceptual model of coping with genetic risk in Huntington’s disease.

**Table 1 biomedicines-13-02498-t001:** Thematic matrix of the studies included in the review.

Author (Year)	Country	Participants	Testing Status	Type of Study	Method	Main Focus	Key Findings	Counselling/Clinical Implications	Notes/Observations
**Costa (2024) [41]**	UK	Families receiving genomic diagnoses (rare diseases, incl. HD)	Mixed: diagnosed/undiagnosed	Qualitative, medical anthropology	Ethnographic fieldwork, interviews	Family experience of genomic diagnoses and re-making of disease categories	Families renegotiate identity and illness when facing uncertain/partial genomic results	Counselling must address diagnostic uncertainty and shifting family identities	Transferable: shows how “undiagnosed/diagnosed” status affects coping with genetic risk, applicable to HD
**Díaz Hernández (2021) [42]**	Spain	Individuals with mental disorder diagnosis	Diagnosed	Qualitative	In-depth interviews	Life changes and meaning after mental disorder diagnosis	Participants reported identity disruption and redefinition of life projects	Counselling should integrate meaning-making processes in post-diagnosis adjustment	Transferable: illustrates psychosocial impact of diagnosis and identity, relevant to genetic risk in HD
**Robert & Klitzman (2012)** [43]	USA	Patients and families undergoing genetic testing (various conditions)	Tested/at-risk	Book (qualitative synthesis)	Interviews, case studies	Confronting fate, family secrets, and genetic identity	Individuals negotiate family secrets, fate vs. autonomy, stigma	Counselling should address secrecy, intergenerational narratives, and stigma	Highly relevant: offers broad conceptual insights on family secrecy and fate in genetic testing
**Navon (2011) [44]**	USA	Sociological analysis of new genomic categories	Not disease-specific	Sociological/STS study	Case analysis of genomic designations	How genetics creates new medical categories beyond phenotype	Shows how genomic designations reframe illness identity	Counselling should consider how labels and categories affect patients’ self-concept	Transferable: relevant for HD as a “genomic designation” with diffuse phenotypic uncertainty
**Etchegary (2009) [45]**	Canada	Individuals living with HD genetic risk	At-risk/tested	Qualitative	Interviews	Coping strategies in HD	Coping included denial, avoidance, faith, active planning, secrecy	Counselling must support flexible, evolving coping strategies	Core HD study: directly relevant to coping with genetic risk
**Quaid et al. (2007) [46]**	USA	39 individuals at risk for HD	At-risk, mixed testing status	Qualitative	Semi-structured interviews	Living at risk, concealing risk, preserving hope	Participants often concealed risk to maintain hope and avoid stigma	Counselling must balance hope with realistic preparation	Very relevant: secrecy and hope as coping mechanisms in HD
**Brewer & McGill (2009a)** [47]	UK	Families with Juvenile HD	Diagnosed juvenile cases	Book chapter (qualitative synthesis)	Case studies, family accounts	Family experiences: diagnosis and early stages of Juvenile HD	Shock, grief, disruption of family life; need for support at diagnosis	Counselling should emphasize early-stage support for families	Relevant: illustrates coping in rare juvenile HD, transferable to genetic risk contexts
**Brewer & McGill (2009b)** [48]	UK	Families with Juvenile HD	Diagnosed juvenile cases	Book chapter (qualitative synthesis)	Case studies, family accounts	Family experiences: later stages of Juvenile HD	Families cope with high burden, social isolation, anticipatory grief	Counselling should address long-term burden and bereavement support	Relevant: highlights coping strategies over disease trajectory, parallels adult HD
**Mahmood et al. (2022) [49]**	Canada (review)	63 studies on people with HD	Mixed (pre-manifest, manifest, caregivers)	Scoping review	Thematic synthesis	Lived experiences of people with HD	Coping strategies include avoidance, resilience, meaning-making, family reliance	Counselling should integrate anticipatory coping and resilience-building	Highly relevant: broad synthesis of coping evidence in HD
**Rubio Vizcaya (2023) [50]**	Argentina	Ethnography of intellectual disabilities (not HD-specific)	Diagnosed disabilities	Ethnographic (master’s thesis)	Observations, interviews	Social production of invisible disabilities	Shows stigma and invisibility of “non-obvious” conditions	Counselling must recognize hidden disabilities and stigma	Transferable: useful for understanding invisibility of pre-manifest HD and social stigma
**Etchegary (2011) [51]**	Canada	Adults living with chronic genetic risk (HD and other conditions)	Mixed: tested and at-risk	Qualitative study	In-depth interviews	Everyday coping with chronic genetic risk	Participants described “putting risk on the back burner” as a way of coping, alternating between avoidance and active engagement	Highlights importance of long-term psychological support and normalization of uncertainty	Relevant for understanding chronic uncertainty and avoidance strategies in HD genetic risk
**Cesanelli & Margulies (2019)** [52]	Argentina	Ethnography of elderly care (not HD-specific)	Not applicable	Ethnographic study	Observations and interviews	“Alzheimerization” of old age and its impact on care	Explores cultural narratives shaping illness experience	Implications for counselling in how cultural framings affect disclosure and support	Included for transferable insights into how social narratives influence coping with neurodegenerative risk
**Oliveira et al. (2020) [53]**	Portugal	Families with HD across generations	Mixed: symptomatic, at-risk, tested	Qualitative family study	Semi-structured interviews, family narratives	Transgenerational management of genetic information	Families build a “puzzle” of HD risk over time, with secrecy and gradual disclosure	Genetic counselling must account for intergenerational secrecy and timing of disclosure	Strongly relevant: shows family-level coping, secrecy, and gradual information management
**Pleutim et al. (2024) [54]**	Brazil	Family caregivers of HD patients	Symptomatic relatives	Qualitative study	Interviews with caregivers	Care practices from caregiver perspective	Caregivers highlight burden, lack of resources, and emotional strategies	Counselling should integrate caregiver strain and coping resources	Relevant for the “coping” theme, showing how families deal with progressive burden of HD
**Andersson et al. (2016) [55]**	Sweden	Individuals tested for HD, long-term follow-up	Predictive tested (carriers and non-carriers)	Qualitative longitudinal study	Interviews with tested individuals	Ethical aspects of predictive testing over time	Long-term experiences include identity changes, family impact, and ethical dilemmas	Counselling must prepare not only for testing but also for long-term adaptation	Core for review: shows long-term coping after predictive testing
**Pakenham et al. (2004) [56]**	Australia	Parents of children with Asperger syndrome	Not HD-specific	Quantitative/qualitative mixed	Surveys and interviews	Meaning-making and benefit finding	Positive reinterpretation and sense-making linked to better coping	Counselling can foster meaning-making strategies	Included for transferable evidence on meaning-making as a coping strategy in genetic conditions
**Etchegary (2006) [57]**	Canada	Families discovering HD in family history	At-risk, untested	Qualitative study	In-depth interviews	Impact of discovering HD in family history	Shock, secrecy, identity changes in family members	Need for anticipatory counselling and sensitive family disclosure strategies	Very relevant: direct link to coping with genetic discovery of HD
**Wieringa et al. (2022) [58]**	UK	10 pre-manifest HD carriers	Tested (pre-symptomatic carriers)	Qualitative study	Interpretative Phenomenological Analysis (IPA)	Pre-manifest experience of living with HD	Participants struggled with uncertainty, identity, and anticipation of symptoms	Counselling should address anticipatory anxiety and identity threats	Highly relevant: focuses on lived coping in pre-manifest HD
**Wexler (2014) [59]**	USA	Review of popular culture depictions of HD	Not applicable	Narrative/historical review	Literature/cultural analysis	Representation of HD in popular culture	HD portrayed as tragic, stigmatizing, fatalistic	Counselling must counteract stigma by contextualizing cultural narratives	Relevant for how stigma in culture affects coping and self-image in HD
**Rasmussen & Alonso (2002)** [60]	Mexico	Patients and families in predictive testing	Mixed (tested and at-risk)	Review/conceptual analysis	Narrative review	Predictive genetic diagnosis and implications	Identified ethical tensions, anxiety, and family impact	Supports counselling models that integrate ethical reflection	Relevant: early perspective on ethical/coping dimensions in HD testing
**Ekkel et al. (2021) [61]**	Netherlands	Outpatients with HD	Symptomatic	Qualitative study	Semi-structured interviews	How patients view their future and prognosis	Ambivalence: some avoid thinking, others plan ahead	Counselling should support realistic but hopeful future orientation	Relevant: provides insight into how patients cope with poor prognosis
**Aguilar-Caro & Oviedo (2018)** [62]	Colombia	Case studies of HD families	Symptomatic and at-risk	Qualitative case study book	Ethnographic/clinical case studies	Coping in families affected by HD	Families use faith, solidarity, and narrative reconstruction	Counselling should incorporate cultural and community resources	Relevant for Latin American context and culturally grounded coping

## Data Availability

No new data were created or analyzed in this study. Data sharing is not applicable to this article.

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
