# Peer review of "Living with Risk, Aging with Uncertainty: A Narrative Review of Health and Genetic Vulnerability in Huntington’s Disease"

_biomedicines, 2025, doi:10.3390/biomedicines13102498_

Round 1

Reviewer 1 Report

Comments and Suggestions for Authors

Because this is a narrative review, the selection of papers may be biased, considering only the authors' inferences and opinions. A more robust methodology for assessing study eligibility for manuscript synthesis would be beneficial.

Databases such as EMBASE are mandatory for health studies. Their inclusion would be appreciated.

The search descriptors and synonyms are limited, or the process of creating the search string for study sensitivity and specificity should be described. This should be better addressed.

Evaluations and synthesis of results for included studies should be addressed. A more in-depth evaluation would be appreciated.

The topic is interesting and has merit; however, a narrative review does not accurately address the gaps and does not present results with decision-making power.

Reviewer 2 Report

Comments and Suggestions for Authors

This manuscript addresses an important yet under-synthesised question: how individuals at risk of Huntington's disease (HD) cope with anticipatory uncertainty and develop coping strategies prior to the onset of clinical symptoms. By organising the findings into four themes — genetic diagnosis, identity/family dynamics, emotional coping and life plans/advance planning — the paper highlights the psychosocial and ethical dimensions that are often overlooked in biomedical reviews. The topic is both timely and clinically relevant, and the prose is generally clear.

That said, the paper currently reads as a hybrid 'narrative review supported by a systematic search'. While this positioning is reasonable, the manuscript would benefit from stronger methodological transparency, a richer synthesis and clearer evidentiary scaffolding. In particular, there is only one figure, which is sparse, and the citation base is relatively lean for a review spanning 2000–2025, leaving claims under-anchored in places.

Major comments:

  • The manuscript cites a total of 46 references for an 11-page review covering a 25-year period. By review standards, this is modest, especially given the breadth of topics covered, including psychosocial, ethical, reproductive and counselling issues. Several sections make general claims that would merit additional citations, preferably recent ones. Expanding the bibliography would materially strengthen the synthesis's authority.

  • Duplication and formatting issues in the references suggest that the list is still in draft form, for example: 'Zaharias, 2018' appears twice with different numbering; there is stray metadata, such as 'WGROUP:STRING:PUBLICATION'; there are duplicated author names; there are missing or malformed page ranges/DOIs; and there is trailing punctuation. These should be corrected to ensure accuracy and avoid inflating apparent support.

  • Despite newer literature existing, the text sometimes relies on older foundational sources for mechanistic or clinical overviews (e.g. 2007–2011). Consider balancing these with more current reviews and consensus statements, especially regarding definitions of prodromal/premanifest states, counselling models and family-systems interventions.

  • While the four thematic domains are well chosen, the results largely read as narrative commentary with few study-specific details (e.g. 'n out of 22 studies described X', or brief attributed mini-summaries). Including concise details at study level (country, participants, method, key finding) would make the synthesis more cumulative and less impressionistic.

  • The manuscript contains only one figure: a selection flow chart on page 4 (Figure 1). There are no evidence tables or conceptual figures. For a review, this is insufficient to help readers grasp the subject matter at a glance. Please add at least the following:
    Table: Counselling/clinical implications aligned to each theme (e.g. practical takeaways for genetic counsellors, timing and family communication).
    Figure (Conceptual model): A framework linking risk awareness, identity work, coping strategies and planning decisions, with moderators such as culture, family roles, prior caregiving and health system context.
    Etc.

Reviewer 3 Report

Comments and Suggestions for Authors

Huntington's disease is an autosomal dominant, fully penetrant, monogenic disorder. That due to the disease late onset, devastating prognosis and lack of real treatment has a enormous psychological effect to the patient and family members. This review article is a compilation of published work about the emotional toll of the disease. The article is well written and to the point. 

Minor points

a) the sentence : "A repeat count of ≥40 is associated with a fully penetrant disorder; this approach achieves a mutation detection sensitivity of 100% [8]" is badly written. The authors have as a compound sentence that lacks a coordinating conjunction.

b) At the end of the manuscript line 397 the authors have "6. Patents" as a subtitle. This is not correct. 

Round 2

Reviewer 1 Report

Comments and Suggestions for Authors

The authors were able to adjust the manuscript based on the suggestions and addressed the reviewer's concerns regarding some methodological points. The study showed significant improvement, and the authors should be congratulated. The manuscript is suitable and follows the Journal's standards.

Reviewer 2 Report

Comments and Suggestions for Authors

Following significant revisions by the authors, I believe that the current version of the manuscript is suitable for publication in the journal Biomedicines.
